# Avoidable Mortality between Metropolitan and Non-Metropolitan Areas in Korea from 1995 to 2019: A Descriptive Study of Implications for the National Healthcare Policy

**DOI:** 10.3390/ijerph19063475

**Published:** 2022-03-15

**Authors:** Min-Hyeok Choi, Min-Hui Moon, Tae-Ho Yoon

**Affiliations:** 1Department of Preventive and Occupational & Environmental Medicine, Medical College, Pusan National University, Yangsan 50612, Korea; come2mh@pusan.ac.kr; 2Office of Public Healthcare Service, Pusan National University Yangsan Hospital, Yangsan 50612, Korea; mhmoon@pnuyh.co.kr

**Keywords:** avoidable death, preventable death, treatable death, regional inequality, health inequality

## Abstract

This study aims to investigate the trends of avoidable mortality and regional inequality from 1995 to 2019 and to provide evidence for policy effectiveness to address regional health disparities in Korea. Mortality and population data were obtained from the Statistics Korea database. Age-standardized all-cause, avoidable, preventable, and treatable mortality was calculated for each year by sex and region. Changes in mortality trends between metropolitan and non-metropolitan areas were compared with absolute and relative differences. Avoidable mortality decreased by 65.7% (350.5 to 120.2/100,000 persons) in Korea, 64.5% in metropolitan areas, and 65.8% in non-metropolitan areas. The reduction in avoidable mortality was greater in males than in females in both areas. The main causes of death that contribute to the reduction of avoidable mortality are cardiovascular diseases, cancer, and injuries. In preventable mortality, the decrease in non-metropolitan areas (−192.4/100,000 persons) was greater than that in metropolitan areas (−142.7/100,000 persons). However, in treatable mortality, there was no significant difference between the two areas. While inequalities in preventable mortality improved, inequalities in treatable mortality worsened, especially in females. Our findings suggest that regional health disparities can be resolved through a balanced regional development strategy with an ultimate goal of reducing health disparities.

## 1. Introduction

The age-standardized all-cause mortality rate in South Korea (hereafter Korea) in 2019 was 595.1 per 100,000 persons, the ninth lowest among the Organization for Economic Cooperation and Development (OECD) countries, and Korea was the country with the largest decline, down to 44.7% from 1075.6 per 100,000 persons in 2001 [1]. Previous studies have reported that factors related to mortality reduction include the establishment of the national health insurance system, the development of medical technology, and the implementation of preventive management programs centered on public health centers [2,3]. Moreover, in recent reports, there has been a demand for narrowing the gap between socioeconomic and regional health inequality in Korea beyond the reduction of the national mortality rate [3,4,5].

All-cause mortality tends to be affected by factors outside the healthcare service. Therefore, Rustein et al. (1976) developed the concept of avoidable mortality as several diseases at certain ages that should not occur in the presence of timely and effective healthcare [6]. Subsequently, several researchers developed the concept of avoidable mortality [7,8,9]. Avoidable death refers to death from preventable or treatable causes and has been used as an indicator to evaluate the effectiveness of health promotion policies, such as smoking cessation, alcohol consumption, and healthcare systems [9,10]. In addition, avoidable deaths can be used to identify areas with high healthcare needs and reflect differences in healthcare access in health inequality studies [11,12,13,14]. Causes of death included in avoidable deaths are continuously changing in consideration of the characteristics of each country and the development of medical technology. Recently, the United Kingdom and the OECD adapted avoidable death as an indicator of the performance of healthcare systems [10,15].

Several studies have been conducted on the regional distribution or disparities in avoidable mortality in Korea [3,11,16]. Recently, Kim et al. (2021) studied changes in specific avoidable causes of death, such as cancer [11]. Bahk and Jung-Choi (2020) established a study on the differences in life expectancy of avoidable deaths by age group [2]. In addition, a comparative study targeting the European countries and areas such as urban or rural areas within the country was also conducted [9,12,13,17,18,19]. However, to the best of our knowledge, there are no studies on the inequality in avoidable deaths between metropolitan and non-metropolitan areas in Korea.

Korea, which has experienced rapid growth, has also experienced disproportionate regional development, especially between metropolitan and non-metropolitan areas, for several decades. For example, the imbalanced distribution of medical care resources between metropolitan and non-metropolitan areas has not improved [20]. In addition, the level of health behaviors, such as smoking rate and the risk of chronic diseases such as obesity, is higher in non-metropolitan areas [21,22]. To solve these problems, universal health coverage has been established, and regional public health policies have been strengthened over the past 30 years. However, despite the policy efforts to alleviate regional imbalances, the effectiveness is still unclear.

Despite several studies, there are insufficient research results that can be used as evidence for health policies to evaluate regional disparities by measuring the mortality rate that health policies can directly affect. Therefore, studies on health inequality between regions of avoidable death can be used as a valuable basis for establishing and implementing intervention policies to resolve regional health inequality.

This study aims to (1) analyze the trend of change in avoidable mortality in Korea, (2) identify the level of inequality in avoidable death between metropolitan and non-metropolitan areas, and (3) investigate the effect of unbalanced development between metropolitan and non-metropolitan areas on avoidable mortality.

## 2. Materials and Methods

### 2.1. Data Sources and Study Populations

In this study, raw data on the cause of death statistics and mid-year population from 1995 to 2019 (25 years) were used to calculate the mortality rate. Data on the causes of death statistics are provided by Statistics Korea, including information such as residential area, sex, age, and causes of death in all cases by year. The mid-year population is the average value of the population on January 1 of that year and population on January 1 of the following year and is the most used data as the denominator to calculate the mortality rate. Statistics Korea estimates the official mortality rate using these data. The study population for analysis included a total of 6,420,330 deaths (3,546,052 for males and 2,874,278 for females), excluding foreigners from 1995 to 2019.

### 2.2. Avoidable Death and Age Standardized Mortality Rate

Avoidable mortality was classified by the OECD/Eurostat lists of preventable and treatable causes of death in the November 2019 version [15]. The preventable causes of death can be mainly avoided through effective public health and primary prevention interventions (before the onset of diseases/injuries, to reduce incidence), and treatable (or amenable) deaths can be avoided through timely and effective healthcare interventions, including secondary prevention and treatment (after the onset of diseases, to reduce case-fatality). Whether to be included in avoidable death, preventable and treatable death is determined according to the main cause of death according to the 10th revision of the International Statistical Classification of Diseases (ICD-10). In this study, the classification criteria for avoidable, preventable, and treatable deaths were based on the OECD criteria, that were more recent than the criteria of Nolte and McKee or the Office for National Statistics (ONS) [9,10,15]. According to the OECD criteria, avoidable death has two subcategories: preventable and treatable. In addition, the two subcategories do not overlap. Avoidable causes of death are divided into 13 disease groups, including infectious diseases, cancer, diseases of the circulatory system, and injuries, and 96 causes of death. Representatively, the cancer group includes 17 types of cancer, such as stomach (C16), liver (C22), lung (C33–C34), and colorectal cancer (C18–C21), and the group of diseases of the circulatory system includes 7 causes, such as ischemic heart diseases (I20–I25), cerebrovascular diseases (I60–I69), and hypertensive diseases (I10–I13, I15) [15]. Of the deaths from these causes, only the deaths of those aged 0–74 years are counted as avoidable deaths. For example, among all of the cases of stomach cancer (C16), only those aged 0–74 years are included in the preventable death category of avoidable deaths. Cases corresponding to the seven causes of death, such as tuberculosis (A15–A19, B90, J65) in infectious diseases, cervical cancer (C53) in cancer, and aortic aneurism (I71) in diseases of the circulatory system, were divided into two and classified into either preventable or treatable deaths.

To compare the mortality rate between the metropolitan and non-metropolitan areas, the age distribution was considered. In this study, the direct age-standardized mortality rate for all-cause, avoidable, preventable, and treatable deaths, classified according to the OECD criteria for avoidable death, were calculated using the causes of death statistics, mid-year population by sex (total, male, and female), and type of areas (Korea, metropolitan and non-metropolitan areas) from 1995 to 2019. The total population in 2006 by the 5 year-group, the central year from 1995 to 2019, was used as the standard population for calculating the standardized mortality rate.

### 2.3. Inequality between Metropolitan and Non-Metropolitan Areas

To identify regional inequality of mortality, regions were divided into metropolitan (Seoul, Incheon, and Gyeonggi regions), which is the Seoul Capital Area, and non-metropolitan areas (other regions). Proportion of population was similar for type of areas. However, resources, such as the gross regional domestic product (GRDP) per capita and the number of physicians in non-metropolitan areas, are less than those in metropolitan areas [23]. In 2019, the metropolitan area consisted of 77 municipalities (si-gun-gu) with a population consisting of 25,925,799 persons (50.0%), whereas the non-metropolitan area consisted of 179 municipalities with a population of 25,924,062 persons (50.0%). The GRDP per capita (1000 won) in 2019 was 41,890 in metropolitan areas and 33,305 in non-metropolitan areas. Moreover, non-metropolitan areas have fewer medical resources, such as hospitals, doctors, and medical personnel, than metropolitan areas. The number of practicing physicians per 1000 persons in 2019 was 2.60 in metropolitan areas and 2.32 in non-metropolitan areas. In addition, accessibility to medical care services and health levels in non-metropolitan areas are relatively low compared to those in metropolitan areas [20,23,24,25,26]. Conversely, government-owned public health facilities that implement public health policies are even in each municipality.

The pair comparison measurements (absolute difference and relative difference) were used as a tool to measure inequality in mortality between metropolitan and non-metropolitan areas in this study. The absolute difference in mortality is the value of mortality of non-metropolitan areas minus that of metropolitan areas, and the relative difference is calculated using the mortality of non-metropolitan areas divided by that of metropolitan areas.

### 2.4. Statistical Analysis

The mortality, absolute difference, and relative difference (ratio) by death and sex classification were presented as a time series line graph from 1995 to 2019 to present the trend of change between the type of areas. In addition, to identify the effect of the 13 groups of causes and major diseases on the change in mortality and inequality between metropolitan and non-metropolitan areas, the absolute and relative changes in the mortality rate from 1995 to 2019, classified by groups of diseases, were assessed. All statistical analyses were performed using the STATA MP 17.0 (Stata Corporation, College Station, TX, USA).

### 2.5. Ethics Statements

The present study was approved by the Institutional Review Board of Pusan National University Hospital (IRB No. H-1901-019-075).

## 3. Results

### 3.1. Cases and Age Standardized Mortality Rate of Avoidable Deaths

Table 1 shows the age standardized mortality rates (per 100,000 persons) of all-cause and avoidable deaths by cause groups from 1995 to 2019. A total of 6,420,330 died during the study period. The mortality of avoidable deaths was 207.6 per 100,000 persons (42,632,575 cases, 41.0% of all-cause deaths). Deaths from preventable and treatable causes occurred in 1,910,714 cases (72.6% of avoidable deaths) and 721,861.5 cases (27.4% of avoidable deaths), respectively. The mortality rate of preventable deaths (151.1) was higher than that of treatable deaths (56.5). Non-metropolitan areas had higher age standardized all-cause, avoidable, preventable, and treatable mortality rates compared with those of metropolitan areas.

Cancer was the most common cause of avoidable death (64.9), followed by injuries (49.2) and diseases of the circulatory system (42.4). In male subjects, the distribution of mortality by the cause group was similar to that of the total results, whereas in females, cancer, diseases of the circulatory system, and injuries were in descending order in terms of mortality rate. Lung cancer had the highest avoidable mortality rate (17.1), followed by liver cancer (16.6) and stomach cancer (13.5). In females, stomach cancer was the most common cause of avoidable death in cancer (8.0), followed by lung and liver cancers. Among the circulatory system diseases, cerebrovascular diseases (27.2) and ischemic heart diseases (11.3) had the highest avoidable mortality rates. Avoidable deaths due to death transport accidents and intentional self-harm accounted for more than 70% of the total cases of injuries.

### 3.2. Age Standardized Mortality Rate of All-Cause and Avoidable Death by Year

Figure 1 (Appendix A) presents age-standardized all-cause, avoidable, preventable, and treatable mortality rates in metropolitan and non-metropolitan areas from 1995 to 2019 by year. All mortality rates continued to decrease during the study period. The all-cause mortality rate for the total population decreased by 413.5 per 100,000 persons (56.4%) from 733.1 in 1995 to 319.6 in 2019 (see the first row in Figure 1). The absolute decrease in the all-cause mortality rate according to the type of areas was larger in non-metropolitan areas (446.9) than in metropolitan areas (357.3), and the relative decrease was similar (metropolitan 56.8%, non-metropolitan 54.6%). The avoidable mortality decreased by 65.7% in all areas, 64.5% in metropolitan areas, and 65.8% in non-metropolitan areas during the study period, which was larger than the decrease in all-cause mortality (see the second row in Figure 1).

The preventable mortality was reduced by 172.4 (66.8%) in all areas (see the third row in Figure 1). According to the type of areas, the decrease in preventable deaths in non-metropolitan areas (192.4, 67.2%) was greater than that in metropolitan areas (142.7, 64.9%). The mortality rate for treatable causes decreased overall by 57.9 (62.6%), in metropolitan areas by 56.6 (63.5%), and in non-metropolitan areas by 58.7 (61.6%) (see the fourth row in Figure 1). The trend of decreasing mortality rate by sex was similar to that of the total population for both males and females (the second and third columns in Figure 1). All mortalities in males were higher than those in females during the study period. However, the absolute decrease in males was larger than that in females.

### 3.3. Changes of All-Cause and Avoidable Mortality by Cause Groups

Table 2 shows absolute and relative changes of cause-specific avoidable mortality by type of areas from 1995 to 2019. The cause group with the greatest absolute change of mortality in the preventable death was cancer −52.6 per 10,000 persons, followed by injuries (−42.5 per 100,000 persons) and diseases of the circulatory system (−37.1 per 100,000 persons). The causes with the greatest relative change were diseases of the circulatory system (−80.5%) followed by infectious diseases (−77.6%), endocrine, and metabolic diseases (−77.5%) in preventable death. In treatable death, the change of mortality in cancer was the highest (−37.1 per 100,000 persons, −80.0%) compared to other causes, followed by endocrine and metabolic diseases (−7.0 per 10,000 persons, −77.7%) and infectious diseases (−4.5 per 100,000 persons, −70.8%).

Preventable deaths had greater absolute and relative decreases in non-metropolitan areas than in metropolitan areas. However, the relative decline in treatable deaths was greater in metropolitan areas than in non-metropolitan areas. Preventable deaths, such as cancer (stomach, liver, and lung cancer), injuries (transport accidents), infectious diseases (tuberculosis), and alcohol- and drug-related deaths (alcohol specific disorders and poisonings) had greater reductions in non-metropolitan areas than in metropolitan areas. Treatable deaths, such as cancer (colorectal cancer), ischemic heart diseases, cerebrovascular disease in diseases of the circulatory system, and genitourinary system (renal failure) decreased to a degree less than those in the metropolitan areas from 1995 to 2019. In addition, the treatable mortality of pneumonia in non-metropolitan areas increased more than that in metropolitan areas. The results according to sex are shown in Appendix A.

### 3.4. Inequality of Avoidable Mortality Rate between Metropolitan and Non-Metropolitan Areas

Figure 2 (Appendix A) presents the trends of absolute and relative differences of age standardized mortality rates between metropolitan and non-metropolitan areas during 25 years. In all-cause death (see the first row in Figure 2), the absolute disparity according to the type of areas continued to decrease (from 132.00 in 1995 to 42.45 in 2019). The relative difference (ratio) decreased from 1.20 in 1995 to 1.14 in 2019 but remained at the range of 1.13–1.14 from 2010. The disparity of avoidable mortality (the relative difference > 1.18), especially in preventive death (relative difference > 1.21), between the type of areas had larger relative differences than the all-cause death during all year of study (see the first to third row and the second column in Figure 2). Moreover, the relative difference in treatable death increased from 1.07 in 1995 to 1.13 in 2019, especially in 2008.

In terms of sex, the decreases in regional differences in males were larger than those in females during the study period. In preventable death in females, the regional differences were not significantly changed, in particular, the disparity in treatable death continued to increase, the absolute difference from 1.04 to 2.96 per 100,000 persons to the relative difference from 1.01 to 1.12 (see the third row in Figure 2).

## 4. Discussion

Our study is the first to identify and compare the all-cause, avoidable, preventable, and treatable mortality rates between populations in metropolitan and non-metropolitan areas using the lists of OECD avoidable death. In our results, non-metropolitan areas had higher mortality rates than metropolitan areas. Avoidable mortality showed a higher decrease than all-cause mortality, and the reduction of preventable mortality among avoidable deaths had a greater decrease than treatable mortality. While the preventable mortality decline in non-metropolitan areas was larger than that in metropolitan areas, the decrease in treatable mortality in non-metropolitan areas was lower than that in metropolitan areas. Therefore, the inequality of preventable death in the type of areas was reduced during the study period, but the inequality of treatable death has been stagnated or larger, especially in females.

Previous studies have reported that the avoidable death showed a greater reduction compared to all-cause deaths in Korea and other countries [3,4,9,16,27,28,29,30]. The results of this study are similar to those of the previous studies. Although the categories in avoidable deaths and health policy contexts of countries are different, this trend has been raised as causing the development of health promotion programs carried out by public health institutions, such as public health centers and medical technologies, and the improvement of medical accessibility. In Korea, insurance coverage has been continuously expanding since 1989, when the national insurance system was started for all people in Korea [31]. The policy is in the direction of reducing the burden of personal medical expenses due to non-reimbursable services and severe illness, which improves the approach to medical care and reduces avoidable mortality.

The results of our study showed a greater reduction in preventable mortality than in treatable mortality. Moreover, the gap in preventable death between metropolitan and non-metropolitan areas had continued to decrease, especially in males. Preventive deaths have been reported to be influenced by primary prevention policies, such as non-smoking and alcohol reduction programs [9,15]. In Korea, the 1995 National Health Promotion Act was established, and in 1996, the Local Health Act was enacted. Based on these laws, public health centers, which are established in 256 municipalities (si-gun-gu), have been conducting health promotion programs, especially in smoking cessation and alcohol consumption reduction programs [32]. As a result, the age of the standardized smoking rate declined from 66.3% in 1998 to 35.7% in 2018 in males, but slightly increased from 6.5% (1998) to 7.5% (2018) in female. These policies are estimated to have had a greater impact on the improvement of preventable mortality among males than females. Health promotion policies in Korea have two characteristics. First, programs addressing health problems, such as non-smoking or smoking cessation, physical inactivity, nutrition management, and metabolic syndrome prevention and management, provide services based on the regional health status [32]. Based on the results obtained through surveys such as the community health survey, the local governments and public health centers evaluated the health status of the region, and established a target and program in the local health policy to achieve this goal. Second, the public health centers implement health promotion programs; thus, it is possible to improve equality between regions [33]. The reduction of preventable deaths due to causes such as stomach and lung cancer, ischemic heart diseases, cerebrovascular diseases, and the narrowing of inequality in preventable mortality between regions can be evaluated as the effects of these health promotion policies.

The narrowing of the regional inequality of treatable death was not markedly significant. Moreover, recently, inequality levels had increased in both sexes. The inequality of treatable death is influenced by regional material deprivation levels and the unequal distribution of medical resources, such as the number of beds or physicians across the regions [3,9,13]. Through the Korean National Health Insurance System, the policy for universal health coverage contributed to the improvement of personal medical expenses. However, unlike the regionally balanced distribution of the public health centers, the unbalanced distribution of medical care resources between metropolitan and non-metropolitan areas has not improved [20]. In 2020, the number of inpatient care beds in Korea increased from 9.1 per 1000 persons in 2007 to 13.8 per 1000 persons in 2020 [23,34]. During the same period, 7.3 per 1000 persons of inpatient care beds in non-metropolitan areas in 2007 increased to 10.1 in 2020, while inpatient beds in metropolitan areas increased from 10.9 to 17.5. While the number of physicians (medical doctors) per 1000 persons increased from 2.2 to 3.2 in metropolitan areas during the study period, it had increased from 2.0 to 2.8 in non-metropolitan areas. Based on these data, it can be said that there is a regional imbalance in the arrangement of medical personnel, such as doctors and nurses, who can treat diseases. Based on the results of previous studies and our study, the lack of significant improvement in inequality in treatable death might be caused by the regional imbalance in the distribution of the supply of medical resources [3,35].

In terms of disease groups, the mortality for cancer had a greater decrease than for other diseases. Cancer is the leading cause of death in Korea, and deaths due to cancer can be prevented or treated by primary prevention such as smoking cessation, secondary prevention such as cancer screening programs, and tertiary prevention such as appropriate treatment [11]. In Korea, the National Cancer Screening Program for the six major cancers (stomach, liver, colon, breast, cervix of the uterus, and lung) have been produced since 1990. The cancer screening rates in Korea increased from 12.7% in 2002 to 55.6% in 2019 [23]. Moreover, the Korean government has supported 12 regional cancers since 2004. These cancer-related policies have induced the improvement of avoidable deaths due to cancer. However, in our studies, the inequality of treatable death by cancer was confirmed to be intensifying; therefore, it is necessary to further strengthen the support policy for regional cancer centers. The mortality due to diseases of pregnancy, childbirth, and the perinatal period somewhat increased. The average age at which women bore their child changed from 31.3 years old in 2010 to 33.1 years old in 2020 [36]. In a previous study, maternal age was a major risk factor for very preterm birth, low birth weight, and perinatal deaths [37]. The increase in mortality due to diseases of pregnancy, childbirth, and the perinatal period seems to be affected by the increase in maternal age. In addition, it is estimated that the large relative change from 1995 to 2019 due to these causes was also influenced by the small number of total deaths (19,662 cases, 0.3% among all-cause deaths).

This study had several limitations. First, the study did not confirm the change in avoidable death according to age groups, which were affected differently by health policy. Second, the study did not identify the relationship between factors, such as regional economic states and deprivation level, individual socioeconomic level, and avoidable mortality. Third, heterogeneity according to the degree of urbanization (big city, small city, and rural districts) within non-metropolitan or metropolitan areas was not considered. Further studies on the trend by age group, and the relationship between avoidable death and socio-economic factors or urbanization, need to be performed.

## 5. Conclusions

Based on the results of our study, despite the trend of decreasing avoidable deaths, the inequality between metropolitan and non-metropolitan areas has not been resolved. In particular, the public health sector is evenly distributed by region, and the gap is narrowing, whereas in the treatment sector with severe regional imbalance, the gap is increasing. This means that the problem of regional health disparity can be resolved through a balanced regional development strategy with a clear goal of reducing health disparities.

## Figures and Tables

**Figure 1 ijerph-19-03475-f001:**
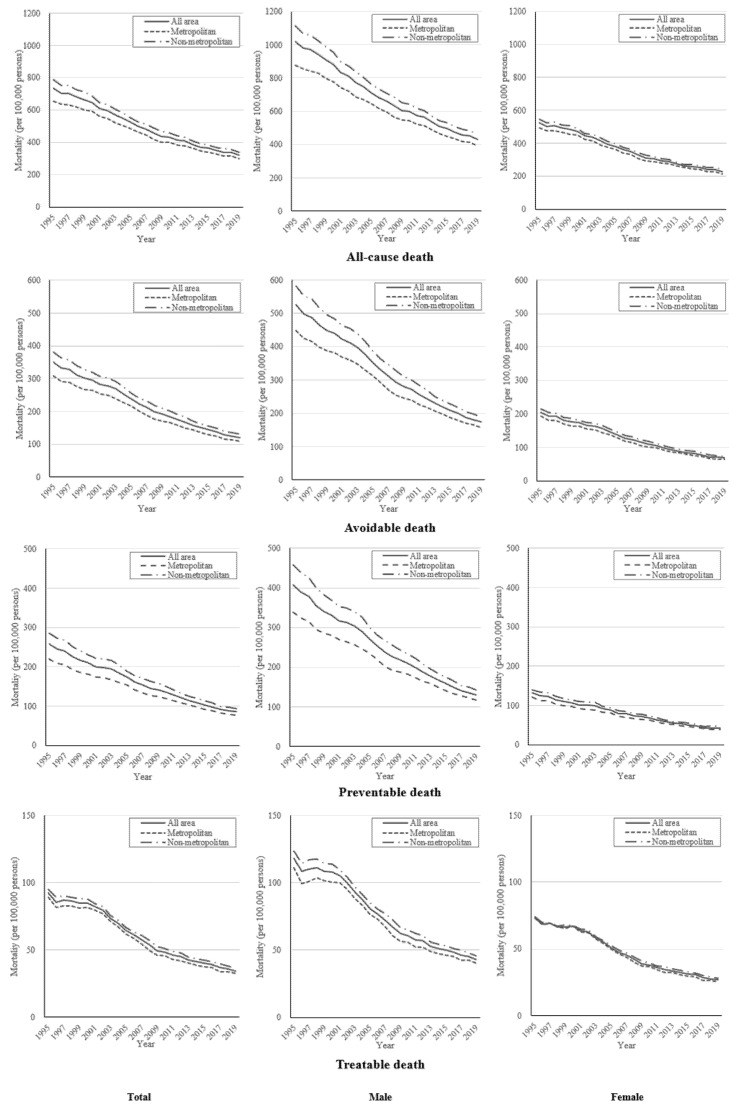
Age standardized all-cause, avoidable, preventable, and treatable mortality rates of metropolitan and non-metropolitan areas by sex from 1995 to 2019.

**Figure 2 ijerph-19-03475-f002:**
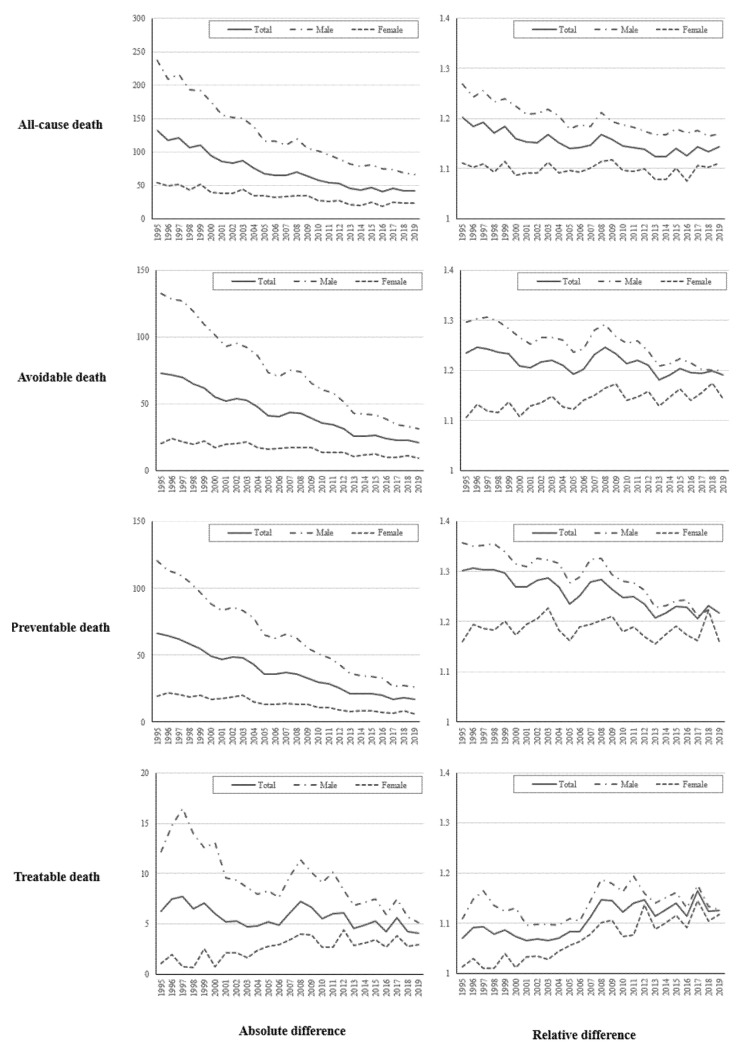
Absolute and relative differences of all-cause, avoidable, preventable, and treatable mortality between metropolitan and non-metropolitan areas from 1995 to 2019.

**Table 1 ijerph-19-03475-t001:** Age standardized all-cause and avoidable mortality rates (per 100,000 persons) according to cause in groups by the type of areas in South Korea from 1995 to 2019.

Cause Group	Total			Male			Female		
	All-Area	Metro ^1^	Non-Metro ^2^	All-Area	Metro ^1^	Non-Metro ^2^	All-Area	Metro ^1^	Non-Metro ^2^
All-cause death (total n = 6,420,330)	468.5	424.4	503.3	636.5	562.5	695.1	336.6	313.8	354.1
Avoidable death (total n = 2,632,575)	207.6	181.5	229.9	303.5	260.3	340.3	120.3	108.9	129.9
Preventable death (total n = 1,910,714) ^3^	151.1	129.2	169.8	233.3	196.6	264.7	75.6	66.8	83.1
Treatable death (total n = 721,861.5) ^3^	56.5	52.2	60.1	70.2	63.8	75.6	44.6	42.1	46.8
Avoidable death by cause group									
Infectious diseases	6.2	5.4	7.0	9.5	8.2	10.6	3.3	2.8	3.6
Tuberculosis	3.3	2.8	3.8	5.6	4.7	6.3	1.3	1.1	1.5
Others	2.9	2.6	3.2	3.9	3.5	4.3	1.9	1.7	2.1
Cancer	64.9	59.0	69.7	95.6	83.9	104.9	38.6	37.2	39.9
Stomach cancer ^4^	13.5	11.8	14.9	19.9	17.2	22.1	8.0	7.0	8.8
Liver cancer ^4^	16.6	14.2	18.6	27.7	23.4	31.2	6.4	5.6	7.0
Lung cancer ^4^	17.1	15.4	18.4	29.0	25.3	31.8	7.2	6.9	7.3
Colorectal cancer ^5^	7.3	7.4	7.2	9.6	9.8	9.5	5.3	5.4	5.2
Breast cancer ^5,6^	-	-	-	-	-	-	5.8	6.3	5.4
Cervical cancer ^6^	-	-	-	-	-	-	2.5	2.5	2.5
Others	6.2	5.7	6.7	9.4	8.3	10.3	3.5	3.4	3.6
Endocrine and metabolic diseases	11.3	10.4	12.0	14.7	13.4	15.8	8.1	7.7	8.5
Diabetes mellitus	11.1	10.3	11.8	14.5	13.2	15.6	7.9	7.5	8.2
Others ^5^	0.2	0.2	0.2	0.1	0.1	0.2	0.2	0.2	0.3
Diseases of the nervous system (epilepsy) ^5^	0.8	0.6	0.9	1.0	0.8	1.2	0.5	0.4	0.7
Diseases of the circulatory system	42.4	38.3	45.7	56.9	51.7	61.2	29.4	26.0	32.0
Ischemic heart diseases	11.3	10.4	12.0	17.2	15.9	18.3	5.8	5.2	6.3
Cerebrovascular diseases	27.2	24.7	29.1	34.8	31.8	37.3	20.4	18.3	22.0
Others	4.0	3.2	4.6	4.9	4.0	5.6	3.1	2.5	3.7
Diseases of the respiratory system	9.5	8.0	10.7	15.4	12.6	17.6	4.7	4.1	5.2
Chronic lower respiratory diseases ^4^	2.7	2.3	3.1	4.8	4.0	5.5	1.1	0.9	1.2
Pneumonia, not elsewhere classified ^5^	3.5	3.1	3.9	5.6	4.8	6.2	1.8	1.6	2.0
Others	3.3	2.6	3.8	5.0	3.8	5.9	1.8	1.6	2.0
Diseases of the digestive system ^5^	1.3	1.0	1.5	1.9	1.5	2.3	0.7	0.6	0.8
Gastric and duodenal ulcer	0.5	0.3	0.6	0.7	0.5	0.9	0.2	0.2	0.3
Others	0.8	0.7	0.9	1.2	1.0	1.4	0.5	0.4	0.5
Diseases of the genitourinary system ^5^	3.5	3.4	3.6	4.5	4.2	4.7	2.7	2.7	2.7
Renal failure	3.3	3.3	3.5	4.3	4.1	4.5	2.6	2.6	2.6
Others	0.2	0.1	0.2	0.2	0.1	0.2	0.1	0.1	0.1
Diseases of pregnancy, childbirth, and perinatal period	1.4	1.3	1.6	1.5	1.4	1.7	1.3	1.2	1.5
Certain conditions originating in the perinatal period ^5^	1.4	1.3	1.6	1.5	1.4	1.7	1.3	1.2	1.4
Others	0.0	0.0	0.0	0.0	0.0	0.0	0.0	0.0	0.0
Congenital malformations	0.7	0.6	0.7	0.7	0.6	0.8	0.6	0.6	0.7
Congenital malformations of the circulatory system ^5^	0.7	0.6	0.7	0.7	0.6	0.7	0.6	0.6	0.7
Others ^4^	0.0	0.0	0.0	0.0	0.0	0.0	0.0	0.0	0.0
Adverse effects of medical and surgical care ^5^	0.1	0.1	0.1	0.1	0.1	0.1	0.1	0.1	0.1
Misadventures to patients during surgical and medical care	0.1	0.1	0.1	0.1	0.1	0.1	0.1	0.1	0.1
Others	0.0	0.0	0.0	0.0	0.0	0.0	0.0	0.0	0.0
Injuries ^4^	49.2	39.8	57.8	73.7	58.8	87.1	25.4	21.3	29.0
Transport accidents	15.5	10.2	20.1	23.8	15.6	31.1	7.2	4.9	9.2
Intentional self-harm	19.7	18.2	21.2	28.0	25.6	30.3	11.7	11.1	12.4
Others	14.0	11.4	16.4	21.9	17.6	25.6	6.4	5.3	7.4
Alcohol-related and drug-related deaths ^4^	16.2	13.5	18.5	28.0	23.0	32.4	4.8	4.3	5.2
Alcohol specific disorders and poisonings	7.7	7.0	8.4	14.3	12.9	15.5	1.4	1.4	1.5
Others	8.4	6.4	10.2	13.7	10.1	16.9	3.3	2.9	3.7

^1^ metro: metropolitan area; ^2^ non-metro: non-metropolitan area; ^3^ preventable death and treatable death are subcategories of avoidable death; ^4^ only included in the category of preventable death; ^5^ only included in the category of treatable death; ^6^ values for only female subjects.

**Table 2 ijerph-19-03475-t002:** Absolute and relative changes of cause-specific avoidable mortality rate according to the type of areas from 1995 to 2019 ^1^.

Cause Group	Absolute Change (per 100,000 Persons)						Relative Change (%)					
	Preventable			Treatable			Preventable			Treatable		
	Total	Metro ^2^	Non-Metro ^3^	Total	Metro ^2^	Non-Metro ^3^	Total	Metro ^2^	Non-Metro ^3^	Total	Metro ^2^	Non-Metro ^3^
All avoidable causes of death	−172.4	−142.7	−192.4	−57.9	−56.6	−58.7	−66.8	−64.9	−67.2	−62.6	−63.5	−61.6
Infectious diseases	−4.0	−3.2	−4.6	−4.5	−4.0	−4.9	−77.6	−75.0	−78.3	−70.8	−71.9	−69.5
Tuberculosis	−4.2	−3.2	−4.9	−4.2	−3.2	−4.9	−91.9	−89.7	−92.9	−91.9	−89.7	−92.9
Others	0.2	0.1	0.3	−0.3	−0.8	0.0	34.1	8.0	56.8	−17.9	−39.7	0.6
Cancer	−52.6	−42.7	−58.5	−1.6	−2.6	−1.0	−63.6	−60.3	−64.7	−12.9	−19.2	−8.8
Stomach cancer ^4^	−22.9	−18.7	−25.4	-	-	-	−80.5	−78.5	−81.1	-	-	-
Liver cancer ^4^	−17.3	−13.5	−19.7	-	-	-	−65.5	−62.1	−66.5	-	-	-
Lung cancer ^4^	−9.3	−8.1	−9.9	-	-	-	−44.4	−42.8	−44.3	-	-	-
Colorectal cancer ^5^	-	-	-	−0.4	−1.1	0.1	-	-	-	−6.0	−16.5	2.4
Breast cancer ^5,6^	-	-	-	1.9	1.5	2.0	-	-	-	41.4	29.0	48.5
Cervical cancer ^6^	−0.5	−0.9	−0.2	−0.5	−0.9	−0.2	−33.7	−51.1	−14.5	−33.7	−51.1	−14.5
Others	−2.8	−1.8	−3.5	−1.8	−1.6	−1.9	−45.9	−34.9	−51.2	−58.0	−55.0	−60.1
Endocrine and metabolic diseases	−6.7	−7.1	−6.5	−7.0	−7.4	−6.8	−77.5	−80.5	−75.2	−77.7	−80.7	−75.3
Diabetes mellitus	−6.7	−7.1	−6.5	−6.7	−7.1	−6.5	−77.5	−80.5	−75.1	−77.5	−80.5	−75.1
Others ^5^	-	-	-	−0.6	−0.2	−0.8	-	-	-	−52.4	−33.3	−61.1
Diseases of the nervous system (epilepsy) ^5^	-	-	-	−0.6	−0.2	−0.8	-	-	-	−52.4	−33.3	−61.1
Diseases of the circulatory system	−37.1	−35.0	−38.5	−37.1	−35.0	−38.4	−80.5	−80.1	−80.5	−80.0	−79.7	−79.9
Ischemic heart diseases	−2.7	−3.0	−2.6	−2.7	−3.0	−2.6	−44.1	−47.3	−42.2	−44.1	−47.3	−42.2
Cerebrovascular diseases	−27.8	−28.4	−27.5	−27.8	−28.4	−27.5	−85.3	−86.2	−84.5	−85.3	−86.2	−84.5
Others	−6.6	−3.6	−8.4	−6.5	−3.6	−8.3	−89.2	−81.4	−91.6	−86.2	−77.9	−88.5
Diseases of the respiratory system	−2.8	−3.6	−2.2	−3.9	−4.0	−3.6	−56.7	−68.8	−46.6	−41.2	−46.9	−35.6
Chronic lower respiratory diseases ^4^	−2.3	−3.2	−1.8	-	-	-	−65.1	−74.7	−56.6	-	-	-
Pneumonia, not elsewhere classified ^5^	-	-	-	1.9	1.1	2.6	-	-	-	63.4	35.9	84.0
Others	−0.4	−0.4	−0.4	−5.8	−5.1	−6.2	−32.9	−41.5	−25.2	−89.2	−91.5	−87.1
Diseases of the digestive system ^5^	-	-	-	−1.7	−1.1	−2.0	-	-	-	−69.1	−62.1	−70.8
Gastric and duodenal ulcer	-	-	-	−1.2	−0.7	−1.5	-	-	-	−84.4	−74.8	−87.5
Others	-	-	-	−0.5	−0.4	−0.5	-	-	-	−47.0	−48.6	−43.8
Diseases of the genitourinary system ^5^	-	-	-	−1.5	−2.5	−0.9	-	-	-	−39.2	−52.8	−27.5
Renal failure	-	-	-	−1.1	−2.1	−0.5	-	-	-	−32.7	−49.5	−17.6
Others	-	-	-	−0.4	−0.3	−0.4	-	-	-	−88.8	−90.1	−87.5
Diseases of pregnancy, childbirth, and perinatal period	0.0	0.0	0.0	0.7	0.8	0.7	0.0	0.0	0.0	207.8	286.1	160.7
Certain conditions originating in the perinatal period^5^	-	-	-	0.7	0.8	0.7	-	-	-	215.6	291.7	168.9
Others	0.0	0.0	0.0	0.0	0.0	0.0	0.0	0.0	0.0	−76.8	−100.0	−69.7
Congenital malformations	0.0	0.0	0.0	−0.8	−0.7	−0.8	28.0	−10.6	104.5	−71.1	−72.5	−69.7
Congenital malformations of the circulatory system ^5^	-	-	-	−0.8	−0.7	−0.8	-	-	-	−71.1	−72.5	−69.7
Others ^4^	0.0	0.0	0.0	-	-	-	28.0	−10.6	104.5	-	-	-
Adverse effects of medical and surgical care ^5^	-	-	-	0.0	0.1	0.0	-	-	-	35.2	180.7	−22.1
Misadventures to patients during surgical and medical care	-	-	-	0.0	0.1	0.0	-	-	-	73.5	580.3	−22.7
Others	-	-	-	0.0	0.0	0.0	-	-	-	−45.9	−76.6	−20.3
Injuries ^4^	−42.5	−32.0	−50.0	-	-	-	−56.9	−53.6	−57.5	-	-	-
Transport accidents	−34.6	−23.7	−42.8	-	-	-	−87.7	−87.4	−87.2	-	-	-
Intentional self−harm	9.6	7.7	11.3	-	-	-	88.0	70.0	100.6	-	-	-
Others	−17.5	−15.9	−18.6	-	-	-	−71.9	−74.3	−69.6	-	-	-
Alcohol-related and drug-related deaths ^4^	−26.7	−19.1	−32.0	-	-	-	−74.8	−69.7	−76.8	-	-	-
Alcohol specific disorders and poisonings	−0.8	−0.2	−1.2	-	-	-	−11.0	−2.5	−14.9	-	-	-
Others	−25.9	−19.0	−30.8	-	-	-	−91.2	−89.9	−91.6	-	-	-

^1^ Absolute change = mortality in 1995—mortality in 2019; relative change (ratio) = (mortality in 1995—mortality in 2019)/mortality in 1995 × 100; ^2^ metro: metropolitan area; ^3^ non-metro: non-metropolitan area; ^4^ only included in the category of preventable death; ^5^ only included in the category of treatable death; ^6^ values for female subjects only.

## Data Availability

All data are available from the Microdata Integrated Service (https://mdis.kostat.go.kr/, accessed on 1 November 2021) and the Korean Statistical Information Service (https://kosis.kr/, accessed on 1 November 2021).

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
