# Peer review of "Avoidable Mortality between Metropolitan and Non-Metropolitan Areas in Korea from 1995 to 2019: A Descriptive Study of Implications for the National Healthcare Policy"

_ijerph, 2022, doi:10.3390/ijerph19063475_

Round 1
Reviewer 1 Report
Avoidable Mortality Between Metropolitan and Non-metropolitan Areas in Korea from 1995 to 2019: Implications for the National Healthcare Policy
Reference IJERPH 1555766
Comments to Authors and Editor
General comments
The manuscript is not currently ready to publish in IJERPH. The study is completely descriptive in nature. Thousands of studies are currently in the literature to explore the trend of mortality whether it is all cause mortality or avoidable mortality or preventable mortality. Everybody knows mortality has been declining in the recent decade due to the advancement of medical technology and public health parameters. I couldn’t find any scientific quality for the study. I strongly request you please revisit the study and change the hypothesis in a significant manner which may benefit for the community and health care system. Find out the causes why such a remarkable dropped off in the recent decade and disseminate to the community, promote to the community.
My few suggestions
The age distribution is influenced by avoidable and non-avoidable mortality, adult mortality can be avoided by preventing diseases using public health parameters, however, the parameters would be different for avoidable child mortality. So worth checking by mortality and age distribution
Also suggest checking mortality for some specific primary causes if data supports. Also some risk factors, for example, smoking, alcohol consumption, morbidity, etc.
Please introduce joint point regression to see the mortality changes between years. Any threshold in the last decade, for example, government introduced any parameters to improve the public health and health sectors which causes a remarkable dropped-off of avoidable mortality.
You can see the mortality was significantly different in the early years between metropolitan and non-metropolitan areas, however reduced the gap in the recent years. Why?
Table 1 is redundant. In the 21st century, nobody is interested to see the total cases. Rate is important which can capture the population in the region.
Author Response
We are sincerely grateful for your thorough review and constructive feedback.
Point 1: The age distribution is influenced by avoidable and non-avoidable mortality, adult mortality can be avoided by preventing diseases using public health parameters, however, the parameters would be different for avoidable child mortality. So worth checking by mortality and age distribution
Response 1: We definitely agree with your opinion. It is necessary to consider the age distribution. The description for considering age distribution was additionally described in the Methods section (page 3, line 117-118). Moreover, we changed Table 1 (from “simple cases of death” to “age-standardized mortality rates”) and complemented the descriptions in the Results section (page 4, line 163-182).
Point 2: Also suggest checking mortality for some specific primary causes if data supports. Some risk factors, for example, smoking, alcohol consumption, morbidity, etc. Please introduce joint point regression to see the mortality changes between years. Any threshold in the last decade, for example, government introduced any parameters to improve the public health and health sectors which causes a remarkable dropped-off of avoidable mortality.
Response 2: Thank you for your suggestion. Our study was not conducted for the purpose of identifying the trend of just mortality. Instead, it was to confirm the difference between metropolitan and non-metropolitan areas. The differences between the two types of areas were compared by calculating the absolute and relative differences in age standardized mortality rates. And Factors, such as the smoking rate and deprivation level, are important variables that affect avoidable mortality. However, There are no indicators of the level of municipalities (si-gun-gu) by year for the study period (1995-2019) in Korea. Our study is the first to identify and compare the all-cause, avoidable, preventable, and treatable mortality rates between metropolitan and non-metropolitan areas in Korea. Although our study has several limitations, it can serve as a basis for establishing health policies in Korea and in other countries. In accordance with your suggestion, we have added the age standardized mortality rate of the major causes of death in the disease group and the relative and absolute change of treatable and preventable mortality over 25 years in Tables 1 and 2 to have more implications for health policy. When related factor data are secured in the future, we plan to conduct further studies on the effects of these factors. Thank you again for your comment.
Point 3: You can see the mortality was significantly different in the early years between metropolitan and non-metropolitan areas, however reduced the gap in the recent years. Why?
Response 3: Thank you for your question. The crude all-cause mortality rates in Korea increased from 532.1 in 1995 to 574.8 in 2019. In contrast, the age standardized mortality rates (ASMR) decreased from 691.9 to 305.4 (approximate -50%). The ASMR is affected by age structure, and the population structure of Korea has rapidly aged over past 25 years. For example, life expectancy at age 0 increased from 73.8 years in 1995 to 83.3 years in 2019. As a result, the absolute difference between the two regions decreased due to a decrease in the overall ASMR. As an indicator to confirm the actual disparity between types of regions, the absolute difference has its limitation. In our study, therefore, the relative differences were calculated and compared additionally: Figure 2 and Table 2 presented the results. Figure 2 (supplementary Tables 7 and 8) and Section 3.4 present the change in absolute and relative differences in mortality rates between metropolitan and non-metropolitan areas. In Table 2, for preventive deaths, the relative differences (ratio) were decreased from 1.30 in 1995 to 1.22 in 2019. Conversely, the relative inequality of treatable deaths increased from 1.07 in 1995 to 1.13 in 2019. The reasons for these results are described in the Discussion section (page 10-11, line 284-325).
Point 4: Table 1 is redundant. Rate is important which can capture the population in the region.
Response 4: Thank you for this comment. We changed Table 1 (from “simple cases of death” to “age-standardized mortality rates”) and complemented the descriptions in the Results section (page 4, line 163-182).
Thank you again for your feedback.

Reviewer 2 Report
The topic is interesting.

Author Response
Point 1: Avoidable Mortality Between Metropolitan and Non-metropolitan Areas in Korea from 1995 to 2019: Implications for the National Healthcare Policy. The authors suggest that theirs is the first to identify and compare all the causes, avoidable, preventable, and treatable mortality rates among metropolitan and non-metropolitan populations in areas using using the lists of OECD avoidable death. The topic is very interesting even if the study does not highlight differences between metropolitan and non-metropolitan areas on avoidable and preventable mortality. The authors rightly point out the potential confounders such as the level of deprivation, socioeconomic level, within the limits. They also consider the analysis to be carried out by age group to be relevant. Indeed, it would be interesting to add some information related to these factors between the two areas under analysis in the background.
Response 1: Thank you for this comment. We have added a description of the regional resources affecting the inequality of avoidable deaths in the Methods section (page 3, line 126-141).
We are sincerely grateful for your considerate review and kind feedback

Reviewer 3 Report
<Attached>
Review comments on the manuscript “Avoidable Mortality Between Metropolitan and Non-metropolitan Areas in Korea from 1995 to 2019: Implications for the National Healthcare Policy”
Overall comments:
This is a well-constructed piece of paper – I would like to first congratulate the authors for their work. This study shows the reduction in avoidable mortality, especially in preventable deaths, both in metropolitan and non-metropolitan areas, from 1995 to 2019. However, it also highlights the gap in treatable deaths between the two regions, especially in female mortality. The authors have nicely discussed these findings in the context of Korean health policies and programmes. I only have minor comments to clarify some explanations or expressions.
Minor comments
1. Introduction
While the authors briefly discussed previous findings in the introduction, it is still not very clear what research gap needs to be resolved. Please explicitly state this research gap (e.g., no research on the difference in avoidable mortality between metropolitan and non-metropolitan areas).
2. Method (section 2.2/2.5)
- I presume that avoidable mortality was divided into two broader categories of deaths, which are preventable and treatable mortalities. However, it is not very clear from the text whether the three types of mortality (avoidable, preventable, and treatable) are mutually exclusive three groups, three groups with some overlaps, or subcategories of avoidable mortality. According to Table 1, I guess the latter (subcategories of avoidable mortality) is the case. Please make Section 2.2 clearer.
- In addition, avoidable mortality has also been categorized into 13 groups by cause of death. However, this is also not clear from the text whether avoidable mortality has been categorized into 13 groups (meaning: no other groups left) OR 13 groups are just selective groups from causes of avoidable mortality (meaning: some other groups left).
- It would also be helpful for some readers if the authors provide examples of non-avoidable mortality causes, since 13 disease groups (96 causes) deem quite comprehensive.
- The authors can make the following sentence a bit clearer: “Cases corresponding to seven causes of deaths, such as tuberculosis…, including half of preventable and treatable deaths, were included.”
3. Method (section 2.3)
- Definition of metropolitan areas: The authors listed the following as examples of metropolitan areas - (Seoul, Incheon, and Gyeonggi regions). Does this mean that the authors defined metropolitan areas as the Seoul Capital Area (SCA)? If so, it would be good to add/state this fact in this section briefly.
- If this is the case, it would also be nice to briefly add the imbalanced development between SCA and non-SCA in the introduction (paragraph 4) (i.e., regional development centred around the SCA) with some examples (e.g., population %, GRDP[regional GDP] %, or other healthcare infra).
- It would be better to clarify what levels of municipalities refer to. Please add a Korean term when first reporting municipalities: e.g., 77 municipalities (si-gun-gu)
4. Table 1
- Please add % to the numbers in the brackets or clearly state that these numbers refer to percentages.
- Please make it clear that ‘preventable death’ and ‘treatable death’ are the subcategories of avoidable death (using spacing or symbolling).
- I think it is better to present the percentages (%) of each cause of avoidable deaths (13 groups) out of total avoidable death, rather than all-cause death. This could help readers promptly understand what causes most contribute to avoidable death in each region. The authors can choose whichever best delivers their intention.
5. Discussion
Overall, the discussion is very interesting, but the phrases such as “the deepening of inequality in treatable death” between the two regions sound a bit too strong, given the findings that (1) treatable death is decreasing in both regions (Figure 1) and (2) absolute difference in treatable death between the two regions is slightly decreasing (decreasing for men but increasing for women) (Figure 2). The authors could slightly tone down their statements.
6. Figure 2
- I expected to see a relatively smooth trend (i.e., less fluctuation) in absolute difference, especially in men and in total. I wonder why there is a sharp increase in an absolute difference between 2007 and 2008 for men. Is there any reason to explain this change? Or is this simply due to a figure scale or random error? This question is totally out of personal curiosity – the authors can add some comments only when they believe that this jump is due to exogenous factors. (otherwise, no need to address this issue)
7. Other very minor
- Section 2.1 (address): The term ‘address’ sounds quite private info. Maybe better to change the term to ‘residential area’ or ‘region’?
- Section 3.4 (relative difference): please make it clear that it is the RATIO of mortality between the two regions when first reporting the relative difference.
Author Response
We are sincerely grateful for your thoughtful reviews and feedback.
Point 1: Introduction: While the authors briefly discussed previous findings in the introduction, it is still not very clear what research gap needs to be resolved. Please explicitly state this research gap (e.g., no research on the difference in avoidable mortality between metropolitan and non-metropolitan areas).
Response 1: Thank you for your suggestion. A sentence including the suggested content has been added in the Introduction (page 2, line 58-60).
Point 2: I presume that avoidable mortality was divided into two broader categories of deaths, which are preventable and treatable mortalities. However, it is not very clear from the text whether the three types of mortality (avoidable, preventable, and treatable) are mutually exclusive three groups, three groups with some overlaps, or subcategories of avoidable mortality. Please make it clear that ‘preventable death’ and ‘treatable death’ are the subcategories of avoidable death (using spacing or symbolling).
Response 2: According to the OECD criteria, avoidable death has two subcategories: preventable and treatable. These two categories do not overlap with each other. Sentences and symbols have been added in the Methods section (page 3, line 102-103) and in Table 1 (symbol 3) to clarify this.
Point 3: Avoidable mortality has also been categorized into 13 groups by cause of death. However, this is also not clear from the text whether avoidable mortality has been categorized into 13 groups (meaning: no other groups left) or 13 groups are just selective groups from causes of avoidable mortality (meaning: some other groups left). It would also be helpful for some readers if the authors provide examples of non-avoidable mortality causes, since 13 disease groups (96 causes) deem quite comprehensive.
Response 3: We completely agree with your opinion. Therefore, we have added an example and more description about the criteria for avoidable deaths in the Methods section (page 3, line 102-116)
Point 4: The authors can make the following sentence a bit clearer: “Cases corresponding to seven causes of deaths, such as tuberculosis…, including half of preventable and treatable deaths, were included.”
Response 4: This sentence has been corrected to convey our intended message (page 3, line 112-116)
Point 5: Definition of metropolitan areas: The authors listed the following as examples of metropolitan areas - (Seoul, Incheon, and Gyeonggi regions). Does this mean that the authors defined metropolitan areas as the Seoul Capital Area (SCA)? If so, it would be good to add/state this fact in this section briefly.
Response 5: This sentence has been added in the Methods section (page 3, line 125-127)
Point 6: If this is the case, it would also be nice to briefly add the imbalanced development between SCA and non-SCA in the introduction (paragraph 4) (i.e., regional development centred around the SCA) with some examples (e.g., population %, GRDP[regional GDP] %, or other healthcare infra).
Response 6: Thank you for your suggestion. This sentence has been added in the Methods section (page 3, line 126-141).
Point 7: It would be better to clarify what levels of municipalities refer to. Please add a Korean term when first reporting municipalities: e.g., 77 municipalities (si-gun-gu).
Response 7: This sentence has been added in the Methods section (page 3, line 131).
Point 8: Table 1: Please add % to the numbers in the brackets or clearly state that these numbers refer to percentages.
Response 8: Table 1 has been modified to reflect valuable suggestions from other reviewers (from cases and proportions to the age standardized mortality rate). The death proportions of preventive and treatable deaths have been modified in accordance with your suggestion (page 4, line 165-167).
Point 9: Discussion: Overall, the discussion is very interesting, but the phrases such as “the deepening of inequality in treatable death” between the two regions sound a bit too strong, given the findings that (1) treatable death is decreasing in both regions (Figure 1) and (2) absolute difference in treatable death between the two regions is slightly decreasing (decreasing for men but increasing for women) (Figure 2). The authors could slightly tone down their statements.
Response 9: We agree with your opinion. This sentence has been modified to reflect your suggestions in the Discussion section (page 11, line 323-325).
Point 10: Figure 2: I expected to see a relatively smooth trend (i.e., less fluctuation) in absolute difference, especially in men and in total. I wonder why there is a sharp increase in an absolute difference between 2007 and 2008 for men. Is there any reason to explain this change? Or is this simply due to a figure scale or random error? This question is totally out of personal curiosity – the authors can add some comments only when they believe that this jump is due to exogenous factors. (otherwise, no need to address this issue)
Response 10: In 2008, the global economic crisis occurred. This crisis may have aggravated the inequality. However, the influence of these factors could not be confirmed in the present study. In the future, we will conduct a study to evaluate the influence of variables that may have an impact. This content was added to the limitations in the Discussion section (page 11, line 347-349)
Point 11: minor suggestions: Section 2.1 (address): The term ‘address’ sounds quite private info. Maybe better to change the term to ‘residential area’ or ‘region’?
Response 11: Thank you for your suggestion. “Address” was changed to “residential area” in line 83, page 2.
Point 12: Section 3.4 (relative difference): please make it clear that it is the RATIO of mortality between the two regions when first reporting the relative difference.
Response 12: “Ratio” was added to reflect the reviewer’s suggestions (page 4, line 150; table 2, symbol 1; page 8, line 246).
Thank you again for your opinions and feedback.
